# All-flexible chronoepifluidic nanoplasmonic patch for label-free metabolite profiling in sweat

Jaehun Jeon [1,2], Sangyeon Lee [1], Seongok Chae[3], Joo Hoon Lee[4], Hanjin Kim [1], Eun-Sil Yu[1,2], Hamin Na[1,2], Taejoon Kang [4,5], Hyung-Soon Park [3], Doheon Lee [1] & Ki-Hun Jeong [1,2] ✉

Wearable sensors allow non-invasive monitoring of sweat metabolites, but their reliance on molecular recognition elements limits both physiological coverage and temporal resolution. Here we report an all-flexible chronoepifluidic surface-enhanced Raman spectroscopy (CEP-SERS) patch for label-free and chronometric profiling of sweat metabolites. The CEP-SERS patch integrates plasmonic nanostructures in epifluidic microchannels for chronological sweat sampling and molecular analysis. An ultrathin fluorocarbon nanofilm modulates surface chain mobility to guide low-temperature solid-state dewetting, forming large-area silver nanoislands on a structured flexible substrate. The wearable patch adheres conformally to skin, collects sequential sweat samples, and supports label-free and multiplexed SERS detection of assorted metabolites. Machine learning-assisted quantification of lactate, uric acid, and tyrosine yields robust metabolic profiles in distinct physical activity states. This wearable optofluidic platform refines molecular sweat sensing and expands the potential for individualized phenotyping in proactive and data-driven healthcare.

Metabolic phenotyping is pivotal in precision medicine, revealing individual health traits for personalized interventions[1]. Metabotypes surpass genetic predispositions, offering valuable insights into the diversity of individuals influenced by extrinsic factors such as behavioral patterns, regimens, or gut microbial activity[2,3]. The interrelation of the metabolite reflects precision health status through metabolic pathway alterations induced by physiological changes or disorders[4–8]. In particular, transient metabolic alterations serve as physiological indicators for proactive healthcare. Such metabolic signals facilitate the instant recognition of physiological disorders to promote optimal health outcomes. For instance, postprandial metabolic rates provide crucial information on individual responses to dietary intake, aiding in the development of personalized nutrition plans[9]. In addition, exercise or lifecycle routines can be tailored by monitoring daily activity-induced metabolite fluctuations[10]. However, conventional approaches profile static metabolites in blood[11,12] or urine[13–15], which pose challenges for dynamic metabolic profiling.

Recent biosensing wearables monitor the metabolic kinetics in biological fluids such as tear fluid[16,17], saliva[18], or sweat[19,20]. Sweat, unlike others, exhibits chemical abundance[21,22] and simple sample collection with less contamination[23], thus allowing in-situ profiling of metabolite alteration in proactive healthcare[24]. For example, analyzing lactate dynamics informs optimal exercise routines by evaluating the lactate threshold or maximal lactate steady state during

[1]Department of Bio and Brain Engineering, Korea Advanced Institute of Science and Technology (KAIST), 291 Daehak-ro, Yuseong-gu, Daejeon 34141, Republic of Korea. [2]KAIST Institute for Health Science and Technology (KIHST), KAIST, 291 Daehak-ro, Yuseong-gu, Daejeon 305-701, Republic of Korea. [3]Department of Mechanical Engineering, Korea Advanced Institute of Science and Technology (KAIST), 291 Daehak-ro, Yuseong-gu, Daejeon 34141, Korea. [4]Bionanotechnology Research Center, Korea Research Institute of Bioscience and Biotechnology (KRIBB), 125 Gwahak-ro, Yuseong-gu, Daejeon 34141, Republic of Korea. [5]School of Pharmacy, Sungkyunkwan University (SKKU), 2066 Seobu-ro, Jangan-gu, Suwon 16419, Republic of Korea. ✉e-mail: kjeong@kaist.ac.kr

physical activities[25–27]. In addition, postprandial changes of branched-chain amino acid or uric acid provide prognostic cues for metabolic syndrome[28–30] or gout[31], respectively. Moreover, cortisol levels[32–34] indicate psychological stress responses during daily life. Furthermore, sweat is readily collected by epidermal interfaces with simple absorbents[35–37], or microfluidic patches[38–40], exhibiting high compatibility with assorted wearable sensors. Such epidermal sweat sensors often utilize electrochemical[26,27,30,31,33,34,36,38,41] or colorimetric methods[42–45]. However, molecular recognition elements such as antibodies or enzymes on active sensing sites still constrain multiplexed and unbiased biomarker detection, hindering full insight into individual physiology.

Surface-enhanced Raman spectroscopy (SERS) allows label-free and quantitative detections of diverse biochemicals[46–49]. In particular, the label-free capability facilitates biomarker discovery by revealing interrelationships among multiple metabolites[50,51]. Threrefore, plasmonic nanostructures are integrated into flexible epidermal patches for metabolic profiling through sweat[48,52–54]. Recent advancements in plasmonic patches combine with epidermal microfluidic channels for not only transient analysis[51–59] but also in-situ collection and chronological profiling of biochemicals[60–63]. In addition, functional microfluidics help advanced sweat analaysis such as direct quantification through consistent sample volume[60,63], incubation for highly sensitive detection[64], and sequential sampling[65–67] for chronological sweat profiling. However, conventional plasmonic nanofabrication on flexible substrates, such as oblique angle deposition[68,69] or pattern transfer[46,70,71], faces technical

challenges. Poor conformity on complex topographies hinders integration with sophisticated functional microfluidics. Consequently, embedding high-performance plasmonic nanostructures onto fully flexible epifluidic platforms remains difficult, particularly for precise, multiplexed, and time-resolved sweat profiling.

Here we report an all-flexible chronoepifluidic SERS patch (CEP-SERS patch) for label-free profiling of sweat metabolites. The CEP-SERS patch consists of a plasmofluidic channel layer (PCL) for sweat collection, storage, and SERS analysis, and a dermal contact layer (DCL) for skin attachment (Fig. 1a). PCL includes a flexible plasmonic SERS substrate and a microfluidic sequential sampler. The SERS substrate features plasmonic nanoislands on ultrathin fluorocarbon-coated PDMS membrane, driven by low-temperature solid-state dewetting of thin silver film. The ultrathin fluorocarbon film effectively dewets the thin metal film directly on the sequential sampler, resulting in plasmonic structures with strong electromagnetic hotspots for highly sensitive SERS analysis. In addition, the sequential sampler serially collects the sweat via the microfluidic channel with capillary bursting valves, which spatially separate the sweat over time. Furthermore, DCL contains medical adhesive with a sweat collection port and interconnects the PCL on the skin for stable sweat collection. Finally, the CEP-SERS patch allows machine-learned label-free quantification of multiple metabolites in chrono-sampled sweat, thus quantitatively profiling sweat over time during physical activities. Such all-flexible feature ensures conformal contact for on-skin chronological sweat collection and label-free quantification of metabolites.

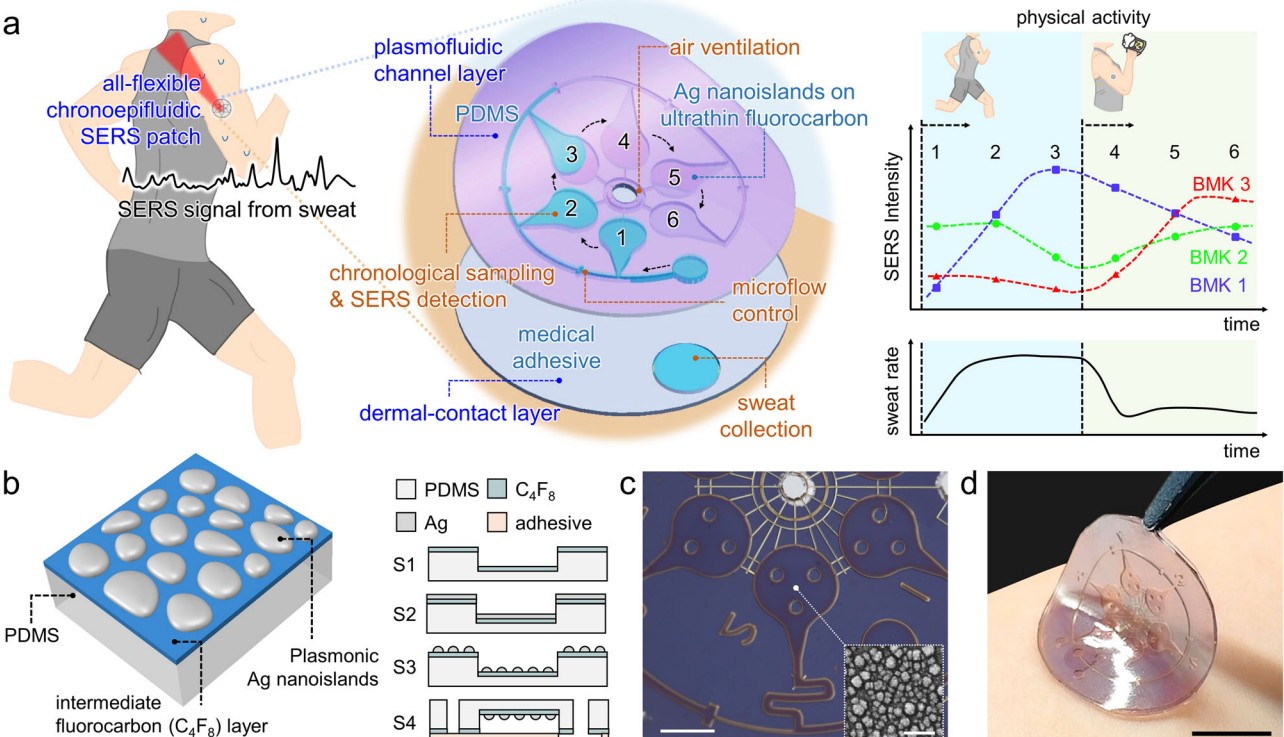

**Fig. 1 | All-flexible chronoepifluidic SERS patch. a** Schematic illustration of an all-flexible chronoepifluidic SERS patch (CEP-SERS patch) comprising plasmofluidic channel layer (PCL) and dermal-contact layer (DCL). PCL includes an all-flexible plasmonic SERS substrate and a chronoepifluidic sweat sampler. The flexible SERS substrate features plasmonic nanoislands on ultrathin fluorocarbon-coated PDMS membrane using low-temperature solid-state dewetting of thin silver film. The sweat sampler sequentially collects the sweat via the microfluidic channel with capillary bursting valves, which spatially separates the sweat over time. DCL interconnects the PCL on the skin for stable sweat collection, which contains medical adhesive with a sweat collection port. The CEP-SERS patch provides

conformal contact on human skin and label-free sweat profiling of diverse metabolites from sequentially sampled sweat. **b** Schematic illustration of micro and nanofabrication for the CEP-SERS patch. The device fabrication includes fluorocarbon coating (S1), thermal evaporation of Ag (S2), low-temperature solid-state dewetting (S3), and microfluidic encapsulation (S4). The ultrathin fluorocarbon film effectively dewets the thin metal film on the sweat sampler, resulting in plasmonic structures with strong electromagnetic hotspots, which facilitate highly sensitive SERS analysis. Optical images of (**c**) the PCL (Scale bar: 1 mm) with an inset SEM image of Ag nanoislands (Scale bar: 100 nm) and (**d**) the CEP-SERS patch on the skin (Scale bar: 10 mm).

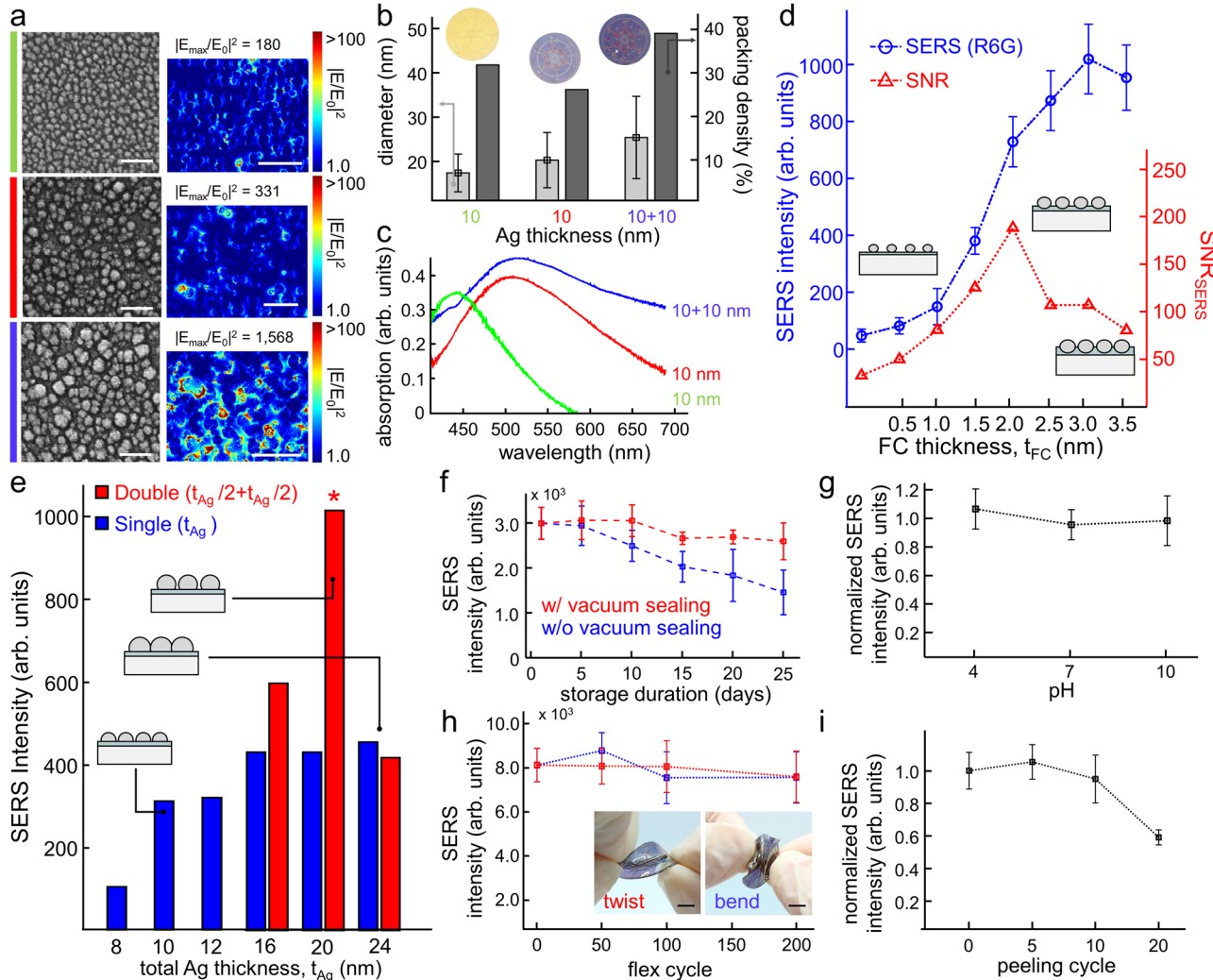

**Fig. 2 | All-flexible SERS Substrate: Ag nanoislands on ultrathin fluorocarbon.**
**a** SEM images (left) and E-field distribution (right) of Ag nanoislands (Scale bar: 100 nm), (**b**) structural features including diameter and surface coverage (calculated from three SEM images taken at different regions of the same sample), and (**c**) absorption spectra for 10 nm Ag film on PDMS without fluorocarbon coating (green), single-dewetted 10 nm Ag film (red), and double-dewetted 10 nm Ag film (blue) on fluorocarbon coated PDMS. **d** SERS intensity of 10 μM R6G at 1365 cm⁻¹ and SNR ($n$ = 12 technical replicates) depending on the fluorocarbon thickness ($t_{FC}$). **e** SERS intensity for 1 μM R6G depending on the Ag film thickness and dewetting repetition (blue: single dewetting, red: repeated dewetting). **f** Long-term stability of

the CEP-SERS patch depending on storage duration measured by SERS peak intensity of 10 μM R6G solution at 1365 cm⁻¹ with (red) and without (blue) vacuum sealing ($n$ = 6 technical replicates). SERS stability (**g**) under different pH measured by SERS peak intensity of 1 μM R6G solution at 1365 cm⁻¹ ($n$ = 6 technical replicates). **h** Mechanical stability measured by SERS peak intensity variation of benzenethiol at 1070 cm⁻¹ after 200 cycles of twisting (red) and bending (blue) (Scale bar: 5 mm). **i** SERS performance stability under repeated peeling cycles of tape measured by SERS intensity of 1 μM R6G at 1365 cm⁻¹ ($n$ = 6 technical replicates). The error bars represent one standard deviation from the mean ($n$ = 6 technical replicates).

The CEP-SERS patch was fabricated by integrating micro- and nanofabricated PCL and DCL (Fig. 1b). First, an ultrathin fluorocarbon layer was coated on the polydimethylsiloxane (PDMS) chronoepifluidic sweat sampler via atmospheric pressure chemical vapor deposition (AP-CVD) (S1). The fluorocarbon coating provides significantly low chain mobility and surface energy, thereby facilitating the low-temperature solid-state dewetting of an Ag thin film. Subsequently, a 10 nm-thick Ag thin film was thermally evaporated onto the fluorocarbon-coated sweat sampler (S2). The evaporated Ag thin film was thermally dewetted at 160 °C for 30 min to form Ag nanoislands (S3). Steps S2 and S3 were repeated to enhance the plasmonic hotspots on the PCL. Finally, an air-venting outlet was punched on the PCL attached to the DCL with a sweat port for encapsulation and skin attachment (S4). Figure 1c, d present optical images of the fabricated PCL, including an inset scanning electron microscopy (SEM) image of the Ag nanoislands, and the CEP-SERS patch on the skin. Details and

stepwise optical images of the fabrication procedure are described in the Supplementary Information (Fig. S1).

## Result

### All-flexible SERS substrate: Ag nanoislands on ultrathin fluorocarbon

The plasmonic properties of Ag nanoislands on the CEP-SERS patch are precisely controlled by the thicknesses of fluorocarbon and Ag thin film. Figure 2a displays SEM images and E-field distribution (Fig. 2a) of Ag nanoislands, calculated by the finite-difference time-domain method (Lumerical FDTD Solutions) method. The structural features (Fig. 2b) and absorption properties (Fig. 2c) are shown for 10 nm Ag film on PDMS without fluorocarbon coating (green), single-dewetted 10 nm Ag film (red), and double-dewetted 10 nm Ag film (blue) on fluorocarbon coated PDMS. The fluorocarbon intermediate layer provides low chain mobility[72–74] and maintains the significant

hydropholicity (Fig. S2), leading to an enlargement of nanoisland diameters on PDMS-based microfluidics by facilitating lower-temperature solid-state dewetting of thin film (Fig. S3). In addition, two-step dewetting further increases the nanoisland diameters and the packing density up to 40%. These morphological modifications induce a red-shift of plasmon resonance (Fig. S4) and enhance the E-field intensity for highly sensitive SERS detection.

The fluorocarbon thickness is determined by signal-to-noise ratio (SNR) in SERS measurements. Figure 2d shows the SERS intensity of 10 μM Rhodamine 6 G (R6G) at 1365 cm$^{-1}$ and SNR depending on the fluorocarbon thickness. A thick fluorocarbon layer enhances the SERS intensity (Fig. S5a) but introduces SERS background noise (Fig. S5b). Therefore, the fluorocarbon thickness was optimized to 2 nm to maximize the SNR, defined as the ratio of the SERS peak intensity at 1365 cm$^{-1}$ to the standard deviation of the fluorocarbon SERS noise signals. Next, the Ag film thickness is optimized for the maximum SERS intensity. Figure 2e shows SERS intensity for 1 μM R6G at 1365 cm$^{-1}$ depending on the Ag film thickness and dewetting repetition. A thick Ag film facilitates the formation of large, densely packed nanoislands during the dewetting process (Fig. S6), leading to enhanced SERS intensity (blue bar). Repeated dewetting further enhances the SERS intensity (red bar) by providing abundant plasmonic hotspots. Note that the coalescence of nanoislands at a total thickness of 24 nm leads to a drastic decline in the SERS signal (Fig. S7). As a result, the CEP-SERS patch is fabricated by a 10 nm repeated dewetting process, which provides the maximum SERS intensity (Fig. S8). The SERS intensity of R6G exhibits strong linearity with concentration (Fig. S9). The SERS substrate provides an average SERS enhancement factor of $1.8 \times 10^7$ with a uniformity of 11.8% (Fig. S10). The CEP-SERS patches are vacuum-sealed for further on-body analysis (Fig. S11) and over 85% of the performance is preserved for up to 25 days through oxidation inhibition (Fig. 2f). The mechanical stability of the CEP-SERS patch is confirmed by measuring the SERS intensity of benzenethiol at 1070 cm$^{-1}$. The chemical stabilities under different pH conditions and chloride ion concentrations are also validated by measuring the SERS intensity of 1 μM R6G at 1365 cm$^{-1}$. No significant SERS signal reduction is observed under varying pH conditions (Fig. 2g). Note that pH stability is evaluated by immersing the substrate in each buffer for 1 h, followed by complete blowing, to avoid interference from the SERS signals of buffer (Fig. S12a). The SERS peaks are decreased by ~40% at NaCl concentrations above 5 mM (Fig. S12b). In addition, all SERS measurements during on-body evaluation are conducted within 12 h, a period during which the signal remained stable within ~10%, to ensure measurement reliability (Fig. S12c). The SERS performance remains stable with no notable degradation after 200 cycles of twisting (red) and bending (blue) (Fig. 2h), and 10 cycles of tape peeling (Fig. 2i). The CEP-SERS patch remains stable and effective for wearable sweat profiling.

## Microfluidic sequential sampler

The sequential sweat sampler allows chronological sample collection through capillary bursting valves (CBVs) in a microfluidic network. Figure 3a illustrates the sampling sequence (left) and SEM images of the microfabricated CBVs (right). The bursting pressure (BP) gradients guide a sample fluid into different chambers for sequential sampling. The BPs are controlled by the width and diverging angle of the CBVs. In particular, the narrow width and the high diverging angle increase the BP in the hydrophobic channel (Fig. S13). The sample fluid enters the first chamber through CBV 1 and fills the chamber. Once the chamber is filled, the fluid then flows into the subsequent chamber through CBV 2. CBV 3 has the highest BP for air ventilation during the sampling. Note that air ventilation allows stable sweat collection without the formation of bubbles. The microfluidic parameters are determined by two-phase fluid dynamic analysis (COMSOL Multiphysics 6.1, Fig. S14). The

sequential loading of distinct colored solutions visualizes the stepwise procedure of chrono-sample collection (Fig. 3b). In addition, the sampling interval is determined by the chamber volume and the flow rate (Fig. 3c). Each chamber sequentially stores a collected sample in a separate space, and the small chamber volume allows dense sampling intervals over time (Fig. S15 and Supplementary Movie 1). In this experiment, the sweat patch with a microchamber volume of 0.5 μL fulfills each chamber within ~2 min at a flow rate of 0.25 μL/min. In addition, the CBVs operate reliably under a flow rate of 5 μL/min (Fig. S16) and a centrifugal acceleration equivalent to 400 rpm (Fig. S17). These conditions reflect physiologically relevant sweat rates and exercise-induced dynamic conditions.

The collected samples are preserved without mixing through on-chip isolation and evaporation suppression. The isolation prevents diffusion between fluids, allowing for long-term storage without mixing (Fig. 3d). An air pocket barrier contains a narrow rectangular structure that traps air via hydrophobicity. Localized evaporation initiates near the structure, leading to sample evaporation from this region. Trapped microbubbles result in physical separation of individual samples (Fig. S18). The effectiveness of isolation is demonstrated by measuring SERS intensity from the deionized water (DI water) in the chamber after sequentially injecting DI water and R6G solution (Fig. 3e). The isolated chambers maintain initial concentration for 30 h (red), while the absence of the air pocket barrier leads to mixing (blue). Low-permeable polyester film, attached to the medical adhesive at the sequential sampler, prevents evaporation. This method retains about 80% of samples after 30 h. Note that evaporation-induced loss is quantified by measuring fluid and air bubble fractions (Fig. S19). This microfluidic sampler facilitates stable sweat collection on the skin and long-term preservation.

## Machine-learned label-free quantification of metabolites

The CEP-SERS patch allows label-free quantification of assorted metabolites. Figure 4a shows the SERS peak intensities of uric acid (red, at 635 cm$^{-1}$), lactate (blue, at 859 cm$^{-1}$), and tyrosine (green, at 1353 cm$^{-1}$) depending on the different concentrations in the range of physiological level (SERS spectra are presented in Fig. S20). A strong linear relationship between SERS peak intensity and concentration facilitates label-free metabolite quantification. The metabolic profiling over time is further shown by tracking the changes in the SERS peak intensity of the target molecules within artificial sweat samples (Fig. S21a). The uric acid concentration is sequentially adjusted to 80 μM, 10 μM, and 20 μM over the injection time by using a syringe pump (Fig. S21b). Figure 4b illustrates the variation in SERS peak intensity of uric acid in chrono-sampled artificial sweat over the collection period (left), utilizing the CEP-SERS patch with chamber volumes of 1.5 μL (blue) and 0.5 μL (red) (SERS spectra are presented in Fig. S21c, d). The CEP-SERS patch with a small chamber volume effectively improves the precise quantification of rapid variations in target molecular concentration by providing dense sampling intervals (right). Chambers with a volume of 0.5 μL are applied for further evaluation to implement dense sampling interval. The CEP-SERS patch contains 17 individual chambers, providing a total maximum sample volume of 8.5 μL.

Machine-learned label-free quantification of metabolites is further conducted by using an autoencoder with a logistic regression model and chemically diverse dataset, which also considers SERS fluctuation (Fig. 3c). SERS signals of target metabolites are affected by surrounding molecular interactions and spectral overlaps (Fig. S22). To improve quantification accuracy and robustness, chemically diverse mixtures are used during model training. A total of 41 different combinations of metabolite concentrations are generated to reflect physiologically relevant background variability (Fig. 3ci and Fig. S23). In addition, SERS signal fluctuations are addressed by collecting 36 spectra per background combination using 10 consecutive 1-s accumulations, and

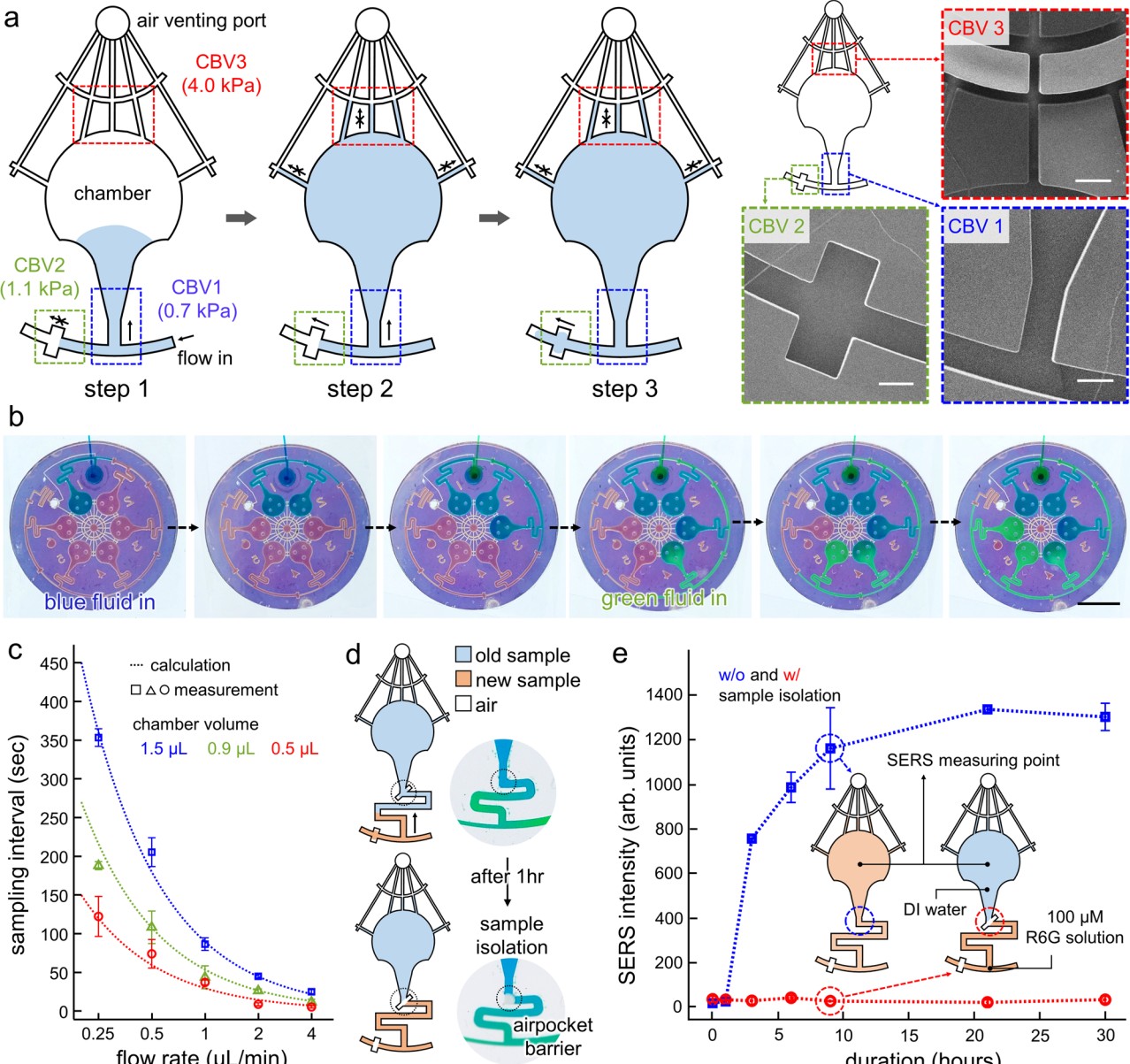

**Fig. 3 | Chrono sample collection and isolation through microfluidic sequential sampler. a** Schematic illustration of sequential sampling (left) and SEM images of capillary bursting valves on the microfluidic sequential sampler (right) (Scale bar: 200 μm). **b** Optical images of sequential sampling of colored dye (Scale bar: 5 mm). **c** Measured sampling interval depending on flow rate and chamber volume ($n = 6$ technical replicates). Each color represents varying chamber volume (blue: 1.5 μL, green: 0.9 μL, red: 0.5 μL). **d** Schematic illustration of on-chip sample isolation utilizing air pocket barrier. **e** The effectiveness of sample isolation is demonstrated by measuring the SERS intensity of diffused R6G at the chamber filled with DI water depending on storage duration with (red) and without (blue) sample isolation. The error bars represent one standard deviation from the mean ($n = 3$ technical replicates).

included in model training to capture signal variability during measurement (Fig. 3cii, Fig. S24)[50]. The autoencoder-based prediction model is then trained by utilizing collected 1476 SERS spectra (Supplementary Data 1) to minimize the total loss function, including the reconstruction and prediction losses (Fig. 3ciii)[75]. The SERS spectra mapped onto the two-dimensional (2D) latent space via autoencoder show that the individual classes are diagonally aligned depending on the concentration of each metabolite (Fig. S25). Figure 4c–e shows the predicted concentrations and the corresponding true values for uric acid ($R^2$: 0.71-0.80), lactate ($R^2$: 0.65-0.83), and tyrosine ($R^2$: 0.82-0.92), which are evaluated with 10 times of repeated random sampling cross-validation (Fig. S26). The prediction errors fall in acceptable ranges. The RMSE is 5.9 μM for uric acid, 3.9 mM for lactate, and 21.9 μM for

tyrosine. These results hold even under chemically diverse backgrounds that emulate physiological variability. The Bland–Altman analysis confirms minimal bias across most concentrations. Slight underestimation is observed at higher levels, likely due to signal saturation in the training set (Fig. S27). The quantification process is explained through the feature extraction calculated by the Shapley additive explanation (SHAP) value (Fig. S28a)[76,77]. The SHAP feature importance is presented with corresponding SERS spectra at 10 mM concentration for uric acid (Fig. S28b), lactate (Fig. S28c), and tyrosine (Fig. S28d). The extracted features reflect key characteristics of the SERS spectra, such as SERS peaks and notches. This model validation demonstrates the machine-learned quantifications are explainable and established based on the molecule-specific SERS signals.

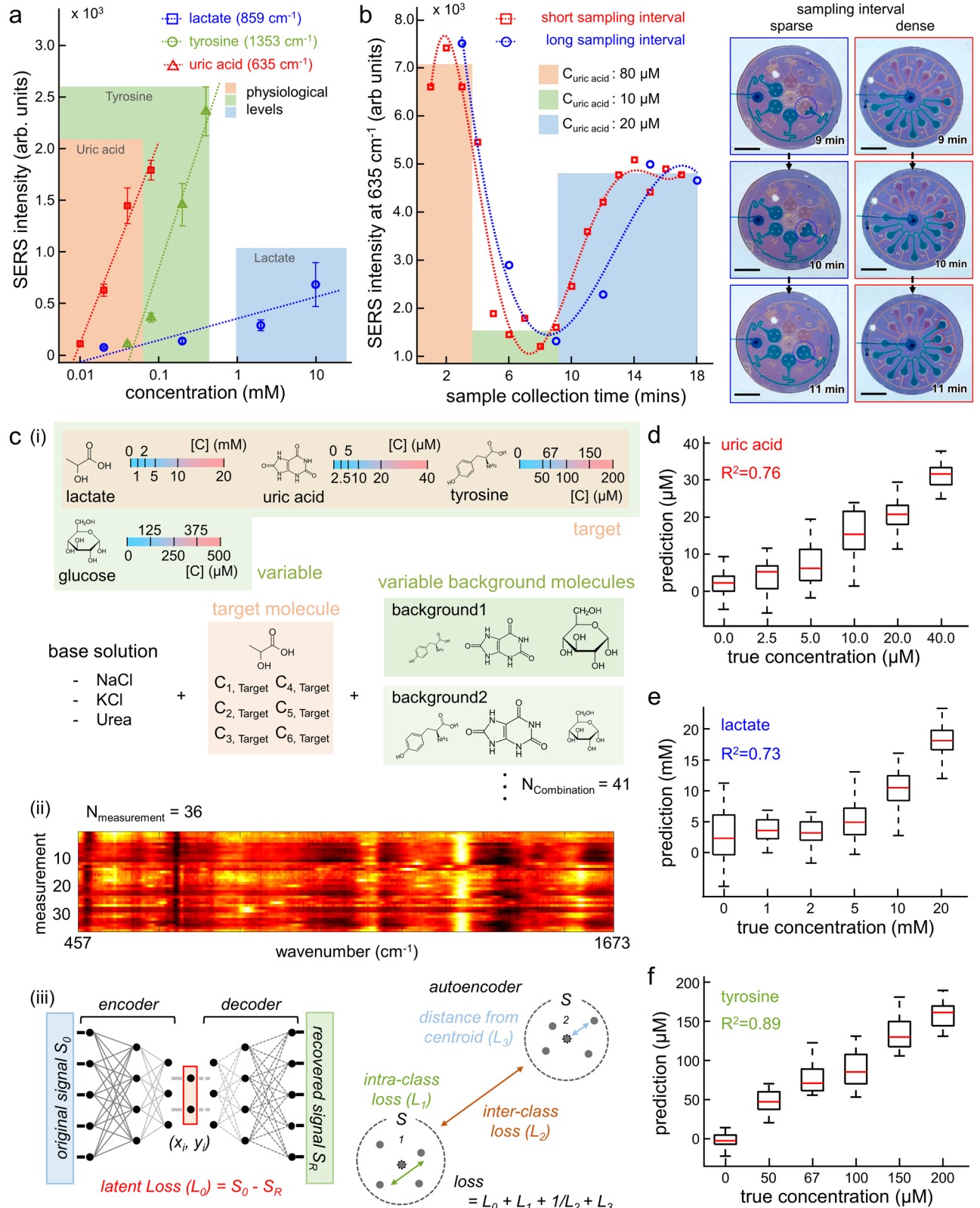

## Label-free human sweat profiling of assorted metabolites

The CEP-SERS patch collects human sweat on the skin and precisely profiles the transient alteration of metabolites without any label. The on-body evaluation includes a treadmill warm-up followed by a climb mill exercise, performed on separate days under fasting conditions and after a purine-rich diet intake (Fig. 5a, Fig. S29). The CEP-SERS patches are attached to multiple sites on the participants' skin to

collect exercise-induced sweat (Fig. 5b). The sweat samples are collected during the evaluation through the CEP-SERS patch. The inset images show the sweat collection using the patch without plasmonic structures to present the sequential sampling. All flexibility of the patch ensures conformal dermal contact, resulting in stable sequential sweat collection (Fig. 5c, top). Note that each participant attaches a total of four patches on both the forehead and shoulder to enhance

**Fig. 4 | Machine-learned label-free quantification of metabolites. a** SERS peak intensity of uric acid (red, at 635 cm⁻¹), lactate (blue, at 859 cm⁻¹), and tyrosine (green, at 1353 cm⁻¹) depending on the concentration ($n = 18$ technical replicates). The error bars represent one standard deviation from the mean. **b** Uric acid profiling in chrono-sampled artificial sweat (left) tracked by SERS intensity at 635 cm⁻¹. Colored bars represent calibrated SERS intensity of input uric acid concentration. Optical images of the sequential sampling over time depending on the chamber volume (blue: 1.5 µL, red: 0.5 µL), which visualize sparse and dense sampling intervals (right). The scale bars represent 5 mm. **c** Schematic of SERS dataset preparation to compensate (i) chemically diverse sweat backgrounds, (ii) signal

fluctuations in SERS measurement, and (iii) machine-learned metabolic quantification using an autoencoder combined with a logistic regression model. Machine-learned quantification of (**d**) uric acid ($n = 75$ for 0 µM, $n = 27$ for 2.5 µM, $n = 29$ for 5 µM, $n = 24$ for 10 µM, $n = 15$ for 20 µM, $n = 19$ for 40 µM, technically replicated), (**e**) lactate ($n = 52$ for 0 mM, $n = 7$ for 1 mM, $n = 10$ for 2 mM, $n = 16$ for 5 mM, $n = 58$ for 10 mM, $n = 42$ for 20 mM, technically replicated), and (**f**) tyrosine ($n = 42$ for 0 µM, $n = 20$ for 50 µM, $n = 7$ for 67 µM, $n = 20$ for 100 µM, $n = 20$ for 150 µM, $n = 8$ for 200 µM, technically replicated), in the mixture presented by prediction depending on true concentration. (center line, median; box limits, upper and lower quartiles; whiskers, 1.5x interquartile range).

sampling diversity. Besides, the CEP-SERS patch securely preserves sweat samples under physical stresses such as compression, twisting, or detachment (Fig. 5c, bottom). Note that the biophysical signals such as heart rate, respiratory exchange ratio (RER), oxygen, and carbon dioxide intake are also concurrently measured during the evaluation to monitor the exercise intensity.

The CEP-SERS patch performs label-free profiling of chrono-sampled human sweat using the machine-learned quantification models for individual metabolites. Sweat samples from four healthy participants are analyzed to assess metabolic differences during exercise with a purine-rich meal versus fasting. The average SERS signals measured from human sweat capture key features of various metabolites in sweat (Fig. S30). SERS signals from chrono-sampled sweat are obtained seven times from each chamber to consider the signal fluctuations (Figs. S31–S34). Note that intrinsic Raman background signals from PDMS and fluorocarbon of the CEP-SERS patch were removed in advance for accurate signal interpretation of metabolites. In addition, human sweat SERS spectra are strictly excluded from the training dataset and used only for independent validation to avoid overfitting. The sweat flow rate for each participant is determined by dividing the total volume of the collected samples by the sampling time (Fig. S35). Label-free uric acid, lactate, and tyrosine quantifications in human sweat are validated using commercial fluorometric assay (FMA) or colorimetric assay (CMA) kits. The sweat is further collected by swiping the microtube to the forehead for FMA and CMA. Machine-learned predictions for uric acid and lactate demonstrate strong agreement with FMA measurements, achieving $R^2$ values of 0.96 and 0.86, respectively (Fig. 5d, e). The Bland–Altman plots show small residuals and consistent agreement within the 95% limits for uric acid (Fig. S36a) and lactate (Fig. S36b). In addition, tyrosine prediction from participant 2–4 shows an error margin of ~ 5 µM to the CMA kit (Fig. 5f). In participant 1, skin lipid contaminants during sampling alter the absorbance, causing a substantial difference of ~ 25 µM from the CMA measurement. The metabolites in the collected sweat samples (Fig. 5g–i) are chronologically profiled with dynamic sweat rate and biophysical signals (Fig. 5j) through the machine-learned SERS quantification models (Chronological profiling results of metabolites, sweat rate, and biophysical signals for four participants under fasting conditions and purine-rich diet intake are presented in Figs. S37–S40). The CEP-SERS patch captures the metabolic alterations associated with the metabolic pathway of each metabolite and dilution. The metabolites in sweat are often diluted over time due to the high sweat rate during exercise[24,31,34,78]. In contrast, lactate shows a temporary increase, reflecting both eccrine sweat gland metabolism[24,26,79] and anaerobic metabolism[80,81] during exercise. The dietary intake increases the overall concentrations of uric acid[31] and tyrosine (amino acid)[30,78,82] in the on-body evaluation (Fig. 5k, l). The metabolic alterations under fasting (blue) and purine-rich diet intake conditions (red) show physiological states for four participants. The averaged alteration (bold red line) reveals that lactate levels remain stable due to minimal dietary influence. In contrast, uric acid and tyrosine increase post-ingestion, reflecting physiological changes in purine metabolism and protein digestion, respectively (Fig. 5m). The

CEP-SERS patch combined with machine-learned metabolic quantification has successfully captured transient physiological changes by label-free and multiplexed detection of exercise- and intake-induced metabolic alterations.

## Discussion

Metabolic profiling offers significant insights into individual physiological states, enhancing the scope of digital phenotyping and advancing personalized healthcare. The wearable sweat sensors capture the transient biochemical alterations during assorted activities and allow on-body metabolic profiling. Furthermore, SERS holds significant potential for unveiling comprehensive physiological information by facilitating label-free universal molecular recognition in sweat. Unlike others (Table S1), the CEP-SERS patch features the integration of nanoplasmonics and functional microfluidics for precise chronological profiling of metabolites. The plasmonic nanostructures are fully integrated into the microfluidic sampler by large-area nanofabrication of Ag nanoislands on fluorocarbon-coated PDMS, enabling both SERS analysis and stable sequential sweat collection. Chrono-sampled SERS spectra are then quantified using a robust machine-learned quantification model trained on diverse background concentration combinations. The CEP-SERS patch has successfully captured the exercise- and intake-induced metabolic changes, demonstrating robust machine-learned multiplexed quantification of sweat metabolites toward dynamic biomarker discovery (Fig. S41). Such unique integration of sequential sampling, label-free analysis, and feature extraction allows biomarker candidate narrowing for diverse physiological conditions. This wearable optofluidic sensor platform advances molecular sweat sensing and offers a new paradigm for individualized phenotyping toward proactive, data-driven healthcare.

## Methods

### Numerical analysis

**E-field distribution.** The electric fields of Ag nanoislands with varying geometries were numerically calculated utilizing a three-dimensional finite-difference time-domain method (Lumerical FDTD Solutions). Geometric parameters of the Ag nanoislands were derived from binary segmented SEM images employing Image J software, with Ag nanoislands modeled as a cylindrical shape.

**Fluid dynamic analysis.** The fluid flow at the capillary bursting valve were numerically calculated by utilizing finite elements methods (FEM, COMSOL Multiphysics 6.1). The calculations were conducted by integrating the two-phase flow and level-set physics modules. In addition, the two-phase flow, level set, and wetted wall phenomena were coupled to derive comprehensive computational results. A time-dependent solver was utilized to compute the temporal evolution of the sequential sampling.

### Materials and reagents

Rhodamine 6 G (R6G, R4127-25G, dye content ~95%), benzenethiol (W361607-SAMPLE-K, ≥98%), sodium L-lactate (L7022-5G, ~98%), urea (U5378-100G), tyrosine (PHR1097-1G, certified reference material),

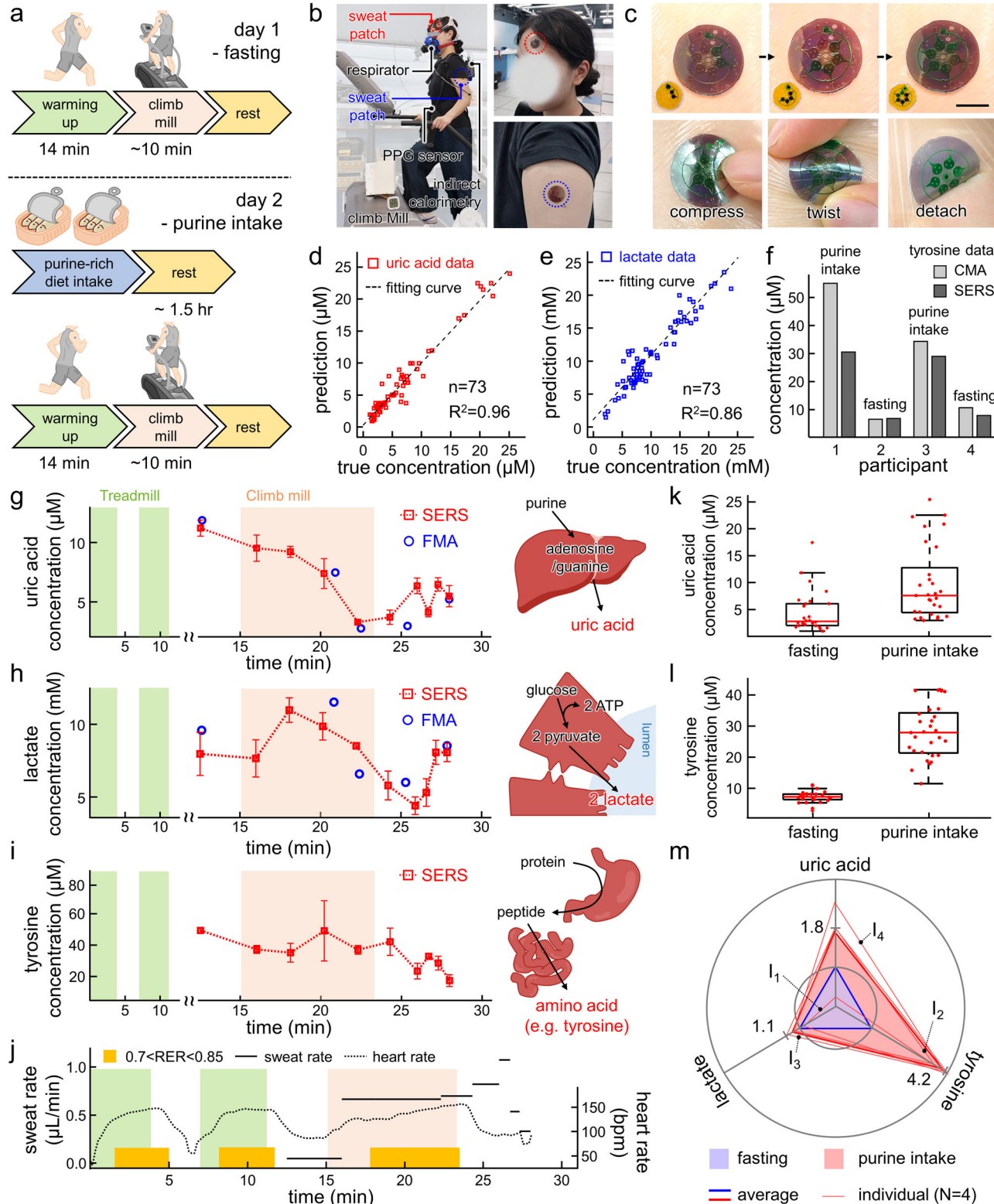

**Fig. 5 | Label-free human sweat profiling of assorted metabolites. a** Schematic illustration of on-body evaluation. Optical images of (**b**) on-body evaluation (left) and the CEP-SERS patch attached to human skin (right), (**c**) sequential sweat sampling (top), and sample preservation under different physical distortions (bottom). The scale bar represents 10 mm. Accuracy of label-free quantification for (**d**) uric acid and (**e**) lactate in the human sweat. **f** Comparison between tyrosine concentrations depending on participants measured by CMA (light gray) and SERS (dark gray). **g**–**i** Chronological profiling (n = 6 technical replicates) of each metabolite during exercise (left) measured by SERS (red rectangles) and FMA (blue

circles) presented with schematic illustrations of each metabolic pathway (right). The error bars represent one standard deviation from the mean. **j** Measured sweat rate (solid line), heart rate (dotted line), and respiratory exchange ratio (RER, yellow bar) during on-body evaluation. Concentration comparison of (**k**) uric acid and (**l**) tyrosine under different physiological conditions. (n = 26 for fasting, n = 29 for purine intake, center line, median; box limits, upper and lower quartiles; whiskers, 1.5x interquartile range; cross, outlier; point, data) (**m**) Relative metabolic phenotyping of individuals ($I_N$, thin solid line) and averaged result (bold solid line) under fasting conditions (blue) and after purine-rich diet intake (red).

glycine (G7126-100G, ≥99%), L-alanine (A7469-25G, ≥98.5%), L-glutamic acid monosodium salt monohydrate (49621-250 G, ≥98.0%), uric acid (U2625-25G, ≥99%), L-ascorbic acid (A5960-25G, ≥99.0%), D-(+)-glucose (G8270-100G, ≥99.5%), creatinine (PHR1462-1G, certified reference material), sodium chloride (S7653-250G, ≥99.5%), and potassium chloride (P5405-250G, ≥99.0%) are purchased from Sigma Aldrich. Medical adhesives were purchased from 3 M (Tegaderm 1622 W, 1522).

## Nanofabrication of CEP-SERS patch
A Si wafer was immersed in buffered oxide etchant for 30 s to remove the native oxide layer. Photoresist (SU-8 2000.5, MicroChem Corp.) was then spin-coated to a thickness of 500 nm to form an adhesion layer. An additional photoresist (SU-8 2100, MicroChem Corp.) was applied to fabricate a microfluidic channel mold with a thickness of 200 μm. A 10:1 mixture of PDMS base and curing agent (Sylgard 184, Dow Corning Corp.) was spin-coated onto the microfluidic channel mold at 100 rpm for 30 s and then at 160 rpm for 30 s. The PDMS was cured on a hotplate at 80 °C for 1 h and 30 min. The PDMS layer was peeled off from the mold after curing and washed in isopropyl alcohol. The fluorocarbon layer was coated on the microfluidic channel via AP-CVD with parameters set to 150 W plasma power, a helium flow rate of 5.0 L/min, and a fluorocarbon flow rate of 2 sccm (IHP-1000, APP Korea). Ag film was thermally deposited at a deposition rate of 1.0 Å/s by using a thermal evaporator (SNTEC Inc., Korea) and dewetted on a hotplate at 160 °C for 30 min. The Ag deposition and dewetting steps were repeated twice to create strong plasmonic hotspots. The air ventilation outlet and sweat inlet were punched on PCL and medical adhesive film (Tegaderm 1622 W, 3 M) for DCL, respectively. The two layers were bonded by double-sided medical adhesive (1522, 3 M) to complete the CEP-SERS patch fabrication.

## Characterization and measurement
**Absorption measurement.** The absorption spectra were measured by using a microscopic spectrometer setup comprising an inverted microscope (Axiovert 200 M, Carl Zeiss) integrated with a white light LED lamp (MCWHL5-C4, Thorlabs Inc.) and a spectrometer (MicroSpec 2300i) featuring a charge-coupled device (CCD) camera (Model PIXIS: 400BR, Princeton Instruments). Absorption spectra of Ag nanoislands on fluorocarbon-coated PDMS were acquired utilizing a 20× objective lens (NA = 0.5).

**SERS measurement.** A helium-neon laser (HRP050, Thorlabs Inc.) operating at a wavelength of 633 nm was utilized in conjunction with a spectrometer featuring a CCD camera, both integrated with an inverted microscope. Light excitation and collection were performed via a 20× objective lens (NA = 0.5). The excitation laser was operated at a power of 5 mW, while data acquisition times were set to 1 s for characterization and extended to 10 consecutive 1 s accumulations for metabolites and human sweat sample measurements. Note that a single SERS spectrum from human sweat was acquired by averaging 6 times of the measurement to minimize signal variances. All the SERS spectra were measured as solution state, and an equivalent volume of DI water was applied to the sample before detection for dry samples.

## Machine-learned metabolic quantification
**SERS data preparation.** First, 1,476 spectra with 41 different concentration combinations including target molecules were provided for machine-learned quantification. The measured SERS spectra have a vector length of 1321, which contains the signal intensities for the Raman shift range of 457–1674 cm$^{-1}$. Each vector was normalized to a range of 0-1.

**Metabolic quantification.** The model consists of a symmetric encoder and decoder with four equally sized layers of 1321 nodes, maintaining a consistent number of neurons across all layers, with a latent layer of length 2 between the encoder and decoder. The model was trained with the Adam optimizer, and the hyperparameters were heuristically tuned to optimize performance. Initially, the learning rate, weight decay, and batch size were set to 1e$^{-4}$, 1e$^{-5}$, and 32 respectively. The maximum number of training epochs was set to 150, with early stopping employed. After the 50$^{th}$ epoch, a scheduled weight decay adjustment was applied to promote further model generalization, reducing the learning rate and weight decay to 1e$^{-5}$ and 1e$^{-6}$, respectively. The model predicts the concentration of each metabolite based on the values along the concentration axis in the latent space. Note that three-quarters of the entire SERS spectra data were used for training, with the remainder reserved for validation. In addition, 10 times of repeated random sampling cross-validation ensures robust model evaluation.

**SERS feature importance calculation.** SHAP (SHapley Additive exPlanations) values were utilized to identify the significant features for concentration prediction. The SHAP calculations were performed using only the encoder part of the model, focusing on the feature importance independent of the increase or decrease in concentration. For each sample, the SHAP values were squared and then averaged across all samples to obtain the overall importance of each feature, thereby identifying key spectral features contributing to the model's predictions.

## On-body evaluation
**Protocols.** The participant performed the following exercise protocol on separate days under fasting conditions and after the purine-rich diet intake, which included 125 g to 250 g of sardines. On the days when the purine-rich diet was consumed, the exercise was initiated after a 1.5 h rest period. The participants wear total four CEP-SERS patches attached to the forehead and shoulder, a wearable metabolic system (Cosmed K5, Cosmed) for respiratory gas analysis, and a heart rate sensor (Polar OH1, Polar Electro Oy) for heart rate monitoring. The participant performed a warm-up consisting of two cycles of running on a treadmill at a 7.2 km/hr speed for 4 min, followed by a 3-min rest period. Subsequently, the participant engaged in exercise on a climbmill with progressively increasing intensity until the heart rate reached 80% of the maximum heart rate. The participant completed an additional 5 min rest period after exercise, concluding the protocol. The CEP-SERS patches were then detached from the body for off-body SERS measurement of the collected human sweat using a benchtop Raman spectrometer. Note that sweat samples were additionally collected using a microtube swapped on the forehead to measure the actual concentration of sweat using the fluorometric assay or colorimetric assay kit. Sweat was collected once after the treadmill exercise, at 3–5 min intervals during the climb mill exercise, and once after a 5 min rest period following the completion of all exercise tasks. Quasi-dynamic sweat rate was estimated based on the number of filled chambers and corresponding elapsed time during the on-body evaluation.

**Fluorometric assays for validation of uric acid and lactate quantification.** The concentrations of uric acid and lactate in collected sweat were validated by using a commercial uric acid assay kit (MAK077-1KT, Sigma Aldrich) and a lactate assay kit (BM-LAC-100, PicoSens™). For uric acid measurement, 5 μL of sweat sample was mixed with 45 μL of uric acid assay buffer, followed by the addition of 2.17 μL of probe and 2.17 μL of enzyme mix. The mixture was thoroughly mixed by pipetting and incubated for 30 min at 37 °C. For lactate measurement, sweat samples were diluted 200-fold with deionized water. A 10 μL aliquot of the diluted sweat sample was mixed with 40 μL of lactate assay buffer, followed by the addition of 0.43 μL of probe and 2.17 μL of enzyme mix. The mixture was thoroughly mixed by pipetting and incubated for

30 min at room temperature. Fluorescence intensity was measured at excitation and emission wavelengths of 535 nm and 590 nm, respectively. Uric acid and lactate concentrations were calculated using a standard curve of each molecule.

**Colorimetric assays for validation of tyrosine quantification.** The concentrations of tyrosine in collected sweat were validated by using a commercial tyrosine assay kit (MET-5073, Cell Biolabs Inc.). Sweat samples were prepared by adjusting their volume to 50 μL with standard tyrosine solutions. The mixed samples were chilled on ice for 30 min and then centrifuged at 10,000 x g for 15 min at 4 °C. The supernatant was carefully transferred to prevent the pellet from dissolving and subsequently passed through a QIAquick Spin Column (28115) by centrifugation (MICRO 17 TR, Hanil Science Co.) at 5,000 g for 10 min at 4 °C. The filtered samples were then used for tyrosine quantification. The 10x enzyme from the kit was diluted to the assay solution, and 15 μL of each prepared sample and standard tyrosine solution was added to individual wells of a transparent 384-well plate, followed by 15 μL of the diluted enzyme solution to make a total volume of 30 μL per well. The plate was incubated at room temperature for 10 min on a horizontal shaker, and absorbance was measured at 490 nm using a plate reader (CYTATION 5 imaging reader, Agilent BioTek). Tyrosine concentrations were then calculated using a standard curve.

### Ethics

All experiments involving human participants were approved by the Institutional Review Board of KAIST (approval number: KH2024-085). All participants (2 males and 2 females, aged 22–25 years) provided informed written consent for participation and for the publication of identifying information. Participants received monetary compensation for their involvement in the study.

### Reporting summary

Further information on research design is available in the Nature Portfolio Reporting Summary linked to this article.

## Data availability

The authors declare that the main data supporting the findings of this study are available within the article and its Supplementary Information. In particular, the SERS spectra used for model training are provided as Supplementary Data 1, and the SERS spectra from human sweat samples are available as Source Data. Source data are provided with this paper.

## Code availability

The custom codes for machine-learned quantification used in this study are publicly available on GitHub at https://github.com/errory230/Linear-scaling-autoencoder.

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

## Acknowledgements

This work was supported by a grant of the National Research Foundation of Korea (NRF) funded by the Ministry of Science ICT & Future Planning (RS-2025-00523089, 2022M3H4A4085645, RS-2024-00438316, NRF-2022M3A9B6017511), Korean ARPA-H Project through the Korea Health Industry Development Institute (KHIDI) funded by the ministry of Health & Welfare, Republic of Korea (RS-2024-00512384, RS-2024-00512239), the Technology Innovation Program (No. RS-2024-00432381) funded by the Ministry of Trade Industry & Energy (MOTIE, Korea), KRIBB Research Initiative Program (KGM1322511, KGM1032511).

## Author contributions

J.J. and K.-H.J. conceived the idea and designed the projects. K.-H.J. supervised the overall research. J.J. designed, fabricated, characterized, and packaged the all-flexible chronoepifluidic SERS patch and conducted overall experiments. S.L., H.K. and D.L. contributed to the machine-learned quantification of metabolites. J.J., S.C. and H.-S.P. conducted the on-body evaluation of the SERS patch. J.J., J.H.L. and T.K. validated the SERS patch using the commercial fluorometric and colorimetric kit. J.J., K.-H.J., S.L., S.C., J.H.L., H.K., E.-S.Y., H.N., T.K., H.-S.P. and D.L. evaluated the experiments and contributed valuable ideas. J.J. and K.-H.J. wrote the manuscript. All authors discussed the results and commented on the manuscript.

## Competing interests

The authors declare no competing interests.
