## [Transparent Peer Review file · Nature Communications]

All-Flexible Chronoepifluidic Nanoplasmonic Patch for Label-free Metabolite Profiling in Sweat

Corresponding Author: Professor Ki-Hun Jeong

Version 0:

Reviewer comments:

Reviewer #1

(Remarks to the Author)

Thanks to the authors for carefully responding to my comments. Based on the response provided, I still have questions related to the robustness of the patch. Hence, would request for additional clarification.

1. How stable are the structures for long-term? Do the salts in sweat affect the binding of the Ag nano islands with the microchannel? Clarification is needed.
2. Answer to 1.2 is contradictory. It is not clear what do you mean by "physiological condition" Sweat pH ranges 5-6. If you are observing signal reduction at pH 4, I assume at pH 5/6 should also be affected. Hence, a pH based correction factor is needed for precise concentration calculation. The authors should calculate and incorporate this.
3. From R7-It is not clear if the DI water added for background subtraction removed from the patch.

Reviewer #2

(Remarks to the Author)

I have carefully reviewed the revised manuscript and the authors' point-by-point responses. While some concerns regarding signal sensitivity and variability remain only partially addressed, the additional data and expanded discussions have improved the clarity and credibility of the work. As noted in my initial review, the concept of using an air ventilation design to avoid sample mixing is practical targeting the key bottleneck in SERS-based sweat analysis. This is a valuable contribution to the field, and I support its publication in Nature Communications.

Reviewer #3

(Remarks to the Author)

The authors have addressed previous concerns with additional data and clarifications. The results are convincing within the scope of the study. I appreciate the improvements and recommend acceptance.

Author's response for the main concerns from the reviewers:

We sincerely thank the reviewers for their constructive and insightful comments. The authors have revised the manuscript and conducted additional experiments to address concerns regarding the long-term stability of the CEP-SERS patch, the influence of pH and salt conditions on SERS performance, and the background subtraction procedure.

In particular, the authors included new data demonstrating the signal stability of the patch after prolonged immersion in NaCl solution, as well as clarified that signal reduction at pH 4 originates from buffer-induced spectral interference rather than degradation of substrate performance. Additional control experiments were performed to decouple these effects, and the rationale for omitting pH correction has been clearly explained and supported by new figures (Fig. 2g, Fig. S12a–c). Furthermore, the authors clarified the background subtraction process, confirming that DI water was only used on reference substrates and did not come into contact with actual sweat samples.

All relevant data and explanations have been incorporated into the revised manuscript and Supplementary Information.

Author's response for the reviewer's point-by-point comments:

For the 1st Reviewer's comments:

Thanks to the authors for carefully responding to my comments. Based on the response provided, I still have questions related to the robustness of the patch. Hence, would request for additional clarification.

1. How stable are the structures for long-term? Do the salts in sweat affect the binding of the Ag nano islands with the microchannel? Clarification is needed.

2. Answer to 1.2 is contradictory. It is not clear what do you mean by "physiological condition" Sweat pH ranges 5-6. If you are observing signal reduction at pH 4, I assume at pH 5/6 should also be affected. Hence, a pH based correction factor is needed for precise concentration calculation. The authors should calculate and incorporate this.

3. From R7-It is not clear if the DI water added for background subtraction removed from the patch.

Author's point-by-point response for detailed comments:

Comment 1.1) How stable are the structures for long-term? Do the salts in sweat affect the binding of the Ag nano islands with the microchannel? Clarification is needed.

Response 1.1) The authors added the long-term stability analysis of the CEP-SERS patch in NaCl solution. The SERS performance exhibited a gradual decrease over time, with the SERS signal of 1 μM R6G retaining approximately 90% of its initial intensity after 12 hours of storage (Fig. R1). The SERS signal of 1 μM R6G remained within $\sim 10\%$ of the initial intensity after 12 hours of storage. Given the fabrication uniformity of the SERS substrate (11.8%), this level of variation falls within the expected range for reliable signal analysis. Therefore, SERS measurements in on-body evaluations were performed within 12 hours of sweat collection to avoid further degradation. The authors included these experimental results in the Supplementary Information (Fig. S12c).

Figure R1 Long-term stability of the CEP-SERS patch in 50mM NaCl solution measured by SERS intensity of 1 μM R6G.

Page	Original	Author's Correction
6	The chemical stability is also validated by measuring the SERS intensity of 1 μM R6G at 1365 cm^{-1} under varying pH (Fig. S12a) and chloride ion concentrations (Fig. S12b), showing a signal decrease of approximately 20% at pH 4 and 40% at NaCl concentrations exceeding 5 mM.	The SERS peaks are decreased by $\sim 40\%$ at NaCl concentrations above 5 mM (Fig. S12b). In addition, all SERS measurements during on-body evaluation are conducted within 12 hours, a period during which the signal remained stable within $\sim 10\%$, to ensure measurement reliability (Fig. S12c).

Comment 1.2) Answer to 1.2 is contradictory. It is not clear what do you mean by "physiological condition" Sweat pH ranges 5-6. If you are observing signal reduction at pH 4, I assume at pH 5/6 should also be affected. Hence, a pH based correction factor is needed for precise concentration calculation. The authors should calculate and incorporate this.

Response 1.2) The authors thank you for pointing out a potentially confusing aspect of our previous pH stability experiment, and now included the results of additional control experiments to clarify this point. In previous experiment, the observed SERS signal reduction at pH 4 is not due to performance degradation of the SERS substrate, but rather results from competitive interactions between R6G and chemical components present in the pH 4 buffer solution. As such, the initial experiment could not fully decouple the influence of pH on substrate performance from signal interference caused by chemical interactions in the buffer.

The authors conducted a control experiment to isolate this effect, where the substrate was first immersed in to the pH 4 buffer without R6G for 1 hour, then completely blown, and subsequently used for SERS measurement of 1 μM R6G. This experimental approach allows the influence of pH on SERS performance to be decoupled from signal interference originating from the buffer solution itself (Fig. R2a). The measured signal showed no significant decrease compared to that at pH 7 or 11, with only $\sim 10\%$ variation (Fig. R2b), which is within the expected measurement error considering substrate uniformity. In addition, immersion in pH 4 buffer for 1 to 12 hours caused no noticeable degradation in SERS performance (Fig. R2c). The SERS signal degradation at pH 4 in initial experiment is not significantly influenced by pH itself but specific interference from pH 4 buffer constituents. Hence, pH correction was not considered in the validation procedure. The authors added experimental data and discussion (Fig. 2g and Fig. S12a).

Figure R2 (a) Measured SERS Spectra of pH 4 buffer solution (blue), 1 μM R6G in the buffer solution (magenta), and 1 μM R6G after 1 hour immersion in the buffer solution (red). (b) pH stability of CEP-SERS patch under varying pH environment and (c) under varying immersion duration in pH 4 buffer solution evaluated by 1 μM R6G SERS intensity measurement at 1365 cm^{-1} .

Page	Original	Author's Correction
6	The chemical stability is also validated by measuring the SERS intensity of 1 μM R6G at 1365 cm^{-1} under varying pH (Fig. S12a) and chloride ion concentrations (Fig. S12b), showing a signal decrease of approximately 20% at pH 4 and 40% at NaCl concentrations exceeding 5 mM.	The chemical stabilities under different pH conditions and chloride ion concentrations are also validated by measuring the SERS intensity of 1 μM R6G at 1365 cm^{-1} . Note that pH stability is evaluated by immersing the substrate in each buffer for 1 hour, followed by complete blowing, to avoid interference from the SERS signals of buffer (Fig. S12a). The SERS peaks are decreased by $\sim 40\%$ at NaCl concentrations above 5 mM (Fig. S12b).

Comment 1.3) From R7-It is not clear if the DI water added for background subtraction removed from the patch.

Response 1.3) The authors clarify that background SERS signals were collected separately using identical substrates without sweat. Since actual sweat measurements were conducted in a wet state, the intrinsic Raman background signals of the device—originating from PDMS and fluorocarbon materials—were also captured under wet conditions using DI water without any analytes. These spectra were measured in advance and subtracted from all sweat SERS spectra to isolate the signal contributions from sweat metabolites. DI water was not applied to the actual sweat collection patch, and it did not affect the measured sweat signals. This point has been clarified in the revised manuscript.

Page	Original	Author's Correction
10	SERS signals from chrono-sampled sweat are obtained seven times from each chamber to consider the signal fluctuations (Fig. S31 – S34).	SERS signals from chrono-sampled sweat are obtained seven times from each chamber to consider the signal fluctuations (Fig. S31 – S34). Note that intrinsic Raman background signals from PDMS and fluorocarbon of the CEP-SERS patch were removed in advance for accurate signal interpretation of metabolites.

For the 2nd Reviewer's comment:

I have carefully reviewed the revised manuscript and the authors' point-by-point responses. While some concerns regarding signal sensitivity and variability remain only partially addressed, the additional data and expanded discussions have improved the clarity and credibility of the work. As noted in my initial review, the concept of using an air ventilation design to avoid sample mixing is practical targeting the key bottleneck in SERS-based sweat analysis. This is a valuable contribution to the field, and I support its publication in Nature Communications.

Author's response: We sincerely thank the reviewer for carefully evaluating our revised manuscript and detailed responses. We truly appreciate the reviewer's recognition of our efforts to address the concerns regarding SERS spectra analysis. In particular, we have expanded our discussion and added supporting data to clarify the robustness of our approach. We are also grateful for the reviewer's positive comments on our air ventilation design and for considering our work a valuable contribution to the field.

For the 3rd Reviewer's comment:

The authors have addressed previous concerns with additional data and clarifications. The results are convincing within the scope of the study. I appreciate the improvements and recommend acceptance.

Author's response: We sincerely thank the reviewer for the constructive and insightful comments throughout the review process. The detailed feedback greatly helped us to improve the clarity, completeness, and overall quality of the manuscript. We truly appreciate the reviewer's recommendation for acceptance and the recognition of the improvements made.